# Molecular mechanism of a covalent allosteric inhibitor of SUMO E1 activating enzyme

Zongyang Lv[1], Lingmin Yuan[1], James H. Atkison[1], Katelyn M. Williams[1], Ramir Vega[2], E. Hampton Sessions[3], Daniela B. Divlianska[3], Christopher Davies[1], Yuan Chen[2] & Shaun K. Olsen [1]

E1 enzymes activate ubiquitin (Ub) and ubiquitin-like modifiers (Ubls) in the first step of Ub/Ubl conjugation cascades and represent potential targets for therapeutic intervention in cancer and other life-threatening diseases. Here, we report the crystal structure of the E1 enzyme for the Ubl SUMO in complex with a recently discovered and highly specific covalent allosteric inhibitor (COH000). The structure reveals that COH000 targets a cryptic pocket distinct from the active site that is completely buried in all previous SUMO E1 structures and that COH000 binding to SUMO E1 is accompanied by a network of structural changes that altogether lock the enzyme in a previously unobserved inactive conformation. These structural changes include disassembly of the active site and a 180° rotation of the catalytic cysteine-containing SCCH domain, relative to conformational snapshots of SUMO E1 poised to catalyze adenylation. Altogether, our study provides a molecular basis for the inhibitory mechanism of COH000 and its SUMO E1 specificity, and also establishes a framework for potential development of molecules targeting E1 enzymes for other Ubls at a cryptic allosteric site.

[1] Department of Biochemistry & Molecular Biology and Hollings Cancer Center, Medical University of South Carolina, Charleston 29425 SC, USA.
[2] Department of Molecular Medicine, Beckman Research Institute of City of Hope, Duarte 91010 CA, USA. [3] Conrad Prebys Center for Chemical Genomics, Sanford Burnham Prebys Medical Discovery Institute at Lake Nona, Orlando 32827 FL, USA. These authors contributed equally: Zongyang Lv, Lingmin Yuan. Correspondence and requests for materials should be addressed to Y.C. (email: ychen@coh.org) or to S.K.O. (email: olsensk@musc.edu)

Reversible post-translational modification of proteins by ubiquitin (Ub) and ubiquitin-like modifiers (Ubls) serves an essential regulatory mechanism for fundamental cellular processes such as cell cycle progression, DNA damage repair, nucleocytoplasmic transport, transcription, and chromatin remodeling[1]. Conjugation of Ub/Ubls to target proteins proceeds through the sequential interactions and activities of parallel cascades of enzymes that are structurally and mechanistically related[2]. E1 enzymes initiate Ub/Ubl conjugation cascades by catalyzing the adenosine triphosphate (ATP)-dependent activation and transfer of their cognate Ub/Ubl to E2-conjugating enzymes, which then function with an array of E3 ligases to catalyze the formation of an isopeptide bond linking the Ub/Ubl to target proteins[3]. Ub/Ubl conjugation exerts its cellular effects by modulating the stability of target proteins, by altering the ability of a target protein to engage in interactions with other cellular proteins, by antagonizing other lysine-targeted post-translational modifications, and/or by triggering conformational changes in target proteins that alter their function or location[4].

Aberrations in Ub/Ubl modifications are associated with the pathogenesis of a wide range of life-threatening diseases, which has resulted in several Ubl E1 enzymes emerging as attractive targets for the development of small molecule therapeutics. For example, dysregulation of SUMOylation is associated with cancer[5–8], neurodegenerative disorders[9,10], and viral infection[11,12]. Currently, the most successful approach to developing inhibitors of Ubl E1 enzymes is by targeting their ATP-binding sites. The first molecule resulting from this strategy is MLN-4924 (or pevonedistat), which is a highly potent and specific inhibitor of the E1 for the Ubl Nedd8 that demonstrates some efficacy in treating acute myeloid leukemia[13,14]. The success of MLN-4924 provided the framework for the subsequent development of structurally related inhibitors specific for Ub E1 and SUMO E1. An inhibitor specific for Ub E1 (MLN-7243 or TAK-243) demonstrates broad efficacy against solid tumors in preclinical models, and is currently in early stages of clinical development[15,16]. The SUMO E1 inhibitor ML-792 demonstrates selective cytotoxicity in c-Myc-overexpressing cells in preclinical studies[17]. Several natural products have also been identified that inhibit SUMO E1 activity, including ginkgolic acid[18], davidiin[19], tannic acid[20], and kerriamycin B[21]; however, their mechanisms of action are unknown.

SUMO E1 is a modular, multi-domain ~110 kDa heterodimer comprised of SUMO-activating enzyme 1 (Sae1) and 2 (Sae2 or Uba2) subunits that harbor two catalytic activities required for activation of SUMO: adenylation and thioester bond formation[22–25]. The active adenylation domain (AAD) and inactive adenylation domain (IAD) are involved in molecular recognition of SUMO and ATP•Mg, and catalyze adenylation of the C terminus of SUMO during the first step of SUMO activation[26,27]. The catalytic Cys domain is split into first catalytic cysteine half-domain (FCCH) and second catalytic cysteine half-domain (SCCH) with the SCCH domain harboring the catalytic cysteine residue that forms a thioester bond with SUMO in the second step of SUMO activation[26,27]. Although not involved in SUMO activation, the SUMO E1 ubiquitin fold domain (UFD) is required for recruitment of the E2 enzyme and the subsequent transfer of SUMO from the E1 catalytic cysteine to the catalytic cysteine of E2[26,28]. Notably, the E1 enzymes for all Ubls share conserved structural features including the AAD, IAD, and catalytic cysteine residues, and all activate their cognate Ubl via a two-step catalytic mechanism involving sequential adenylation and thioester bond formation [29-38]Recent studies have revealed that both the adenylation and thioester bond formation activities of SUMO E1 are catalyzed at a single location on the enzyme that is reconfigured for catalysis of these distinct chemical reactions

via a network of complementary conformational changes[27]. Following adenylation of SUMO, which occurs with the SCCH domain in an open conformation, E1 contacts to ATP•Mg are released, leading to disassembly of the adenylation active site and a 130-degree closure of the SCCH domain (relative to the open conformation) that brings the E1 catalytic cysteine and other residues required for thioester bond formation into proximity of the SUMO C terminus[27,39]. Disassembly of the adenylation active site and concomitant assembly of the thioester bond formation active site following SUMO adenylation and pyrophosphate release serves as a mechanism to drive the SUMO activation process forward[27]. It is likely that other Ubl E1s undergo similar conformational changes during the course of their catalytic cycles[13,27,40].

Here, we present the 2.45 Å crystal structure of SUMO E1 in complex with a newly identified, highly specific inhibitor of SUMO E1 (COH000). Intriguingly, the structure reveals that COH000 binds to a cryptic pocket distinct from the active site that is completely buried in all prior SUMO E1 structures. When in complex with COH000, SUMO E1 residues essential for adenylation are either disordered or displaced from the active site and the SCCH domain undergoes a 180° rotation that is stabilized by a new network of contacts with the adenylation domains that lock the E1 in an inactive state. Interestingly, COH000 exploits salient mechanistic features of the normal SUMO E1 catalytic cycle, including adenylation active site disassembly and SCCH domain rotation; and local unfolding and conformational changes that accompany COH000 binding serve as a mechanism to allosterically couple the adenylation and SCCH domains. Altogether, our structural and biochemical data reveal the molecular basis by which COH000 specifically inhibits SUMO E1 through a covalent allosteric mechanism. Finally, E1s for other Ubls harbor pockets analogous to the COH000 binding site of SUMO E1, raising the possibility that these sites can be exploited for the development of inhibitors targeting other Ubl pathways.

## Results

**Overall structures of apo SUMO E1 and SUMO E1-COH000 complex.** A previously described high-throughput screen (HTS) campaign of an ~300,000 compound library was conducted through the NIH (National Institutes of Health) Molecular Library Probe Production Center Network program (Pubchem AID 2011). The HTS was designed using time-resolved fluorescence resonance energy transfer (Pubchem AID 2006) and AlphaScreen (Pubchem AID 2018) as the primary and secondary assays. All hits were counter-screened against a ubiquitination assay (Pubchem AID 2658) to eliminate compounds that were not specific to SUMOylation. Subsequent hit-to-lead optimization identified COH000 (Pubchem CID 46835082) as a covalent allosteric inhibitor of SUMO E1 adenylation activity that has no effect on SUMO E2 activity[41]. COH000 has also been demonstrated to induce strong anti-tumor effects in colorectal cancer cells as well as mouse and patient-derived xenograft models[41].

In order to elucidate the molecular mechanism by which COH000 inhibits SUMO E1 activity, we determined crystal structures of SUMO E1 alone (SUMO E1[APO]) and in complex with COH000 (SUMO E1[COH000]). The SUMO E1[APO] and SUMO E1[COH000] structures were resolved to 3.1 Å and 2.45 Å, respectively, and were determined from crystals belonging to the same space group with nearly identical unit cell dimensions (Table 1). The SUMO E1[APO] structure adopts an overall architecture resembling previously determined snapshots of the enzyme in an adenylation active state (Protein Data Bank (PDB): 1Y8R, 1Y8Q, 3KYC)[26,27], with the SCCH domain adopting the open conformation and the adenylation active site fully assembled

**Table 1 Data collection and refinement statistics**

| | SUMO E1$^{APO}$ (PDB:6CWZ) | SUMO E1$^{COH000}$ complex (PDB:6CWY) |
|---|---|---|
| *Data collection*[a] | | |
| Space group | $P2_12_12_1$ | $P2_12_12_1$ |
| Cell dimensions | | |
| $a, b, c$ (Å) | 56.1, 115.4, 173.0 | 56.1, 116.0, 174.1 |
| Resolution (Å) | 50–3.10 (3.21–3.10)[b] | 50–2.45 (2.54–2.45) |
| $R_{merge}$ | 0.114 (0.985) | 0.088 (0.880) |
| $R_{pim}$ | 0.054 (0.498) | 0.042 (0.436) |
| $I / \sigma I$ | 10.4 (1.1) | 15.3 (1.2) |
| $CC_{1/2}$ | (0.645) | (0.635) |
| Completeness (%) | 99.5 (96.9) | 99.8 (99.9) |
| Redundancy | 5.4 (4.7) | 5.2 (4.8) |
| *Refinement* | | |
| Resolution (Å) | 47.10–3.10 (3.21–3.10) | 48.20–2.46 (2.52–2.46) |
| No. of reflections | 21,067 (1485) | 41,802 (2765) |
| $R_{work}/R_{free}$ | 0.225 (0.402)/ 0.258 (0.398) | 0.197 (0.264)/0.237 (0.297) |
| No. of atoms | | |
| Protein | 6422 | 6225 |
| Ligand/ion | 1 | 65 |
| Water | – | 45 |
| *B*-factors (Å$^2$) | | |
| Protein | 111.8 | 76.0 |
| Ligand/ion | 77.5 | 86.5 |
| Water | – | 52.7 |
| R.m.s. deviations | | |
| Bond lengths (Å) | 0.002 | 0.002 |
| Bond angles (°) | 0.460 | 0.474 |

[a]All data sets collected from single crystals
[b]Values in parentheses are for highest resolution shell

(Fig. 1a). Analysis of the overall SUMO E1$^{COH000}$ structure shows COH000 wedged between the AAD and IAD at a location distinct from the active site, and comparison to the SUMO E1$^{APO}$ structure reveals a striking ~180° rotation of the SCCH domain and near complete disassembly of the adenylation active site (Fig. 1a, b and Supplementary Movie 1). Since the COH000 binding site is distinct from the active site, the SUMO E1$^{COH000}$ structure suggests that COH000 inhibits SUMO E1 activity through an allosteric mechanism. As in all other SUMO E1 structures[26,27], the FCCH domain is largely disordered.

Notably, both a significant rotation of the SCCH domain (domain alternation) and adenylation active site disassembly were previously observed in the thioester bond formation active conformation of SUMO E1[27]. In this case, the ~130° closure of the SCCH domain (relative to the open conformation observed in adenylation active SUMO E1 snapshots) transits the catalytic cysteine residue ~34 Å such that it is proximal to the SUMO C terminus where thioester bond formation occurs[27] (Fig. 1a, b). Structural comparison of the two catalytic snapshots shows that disassembly of the adenylation active site is necessary to achieve a SUMO E1 conformation with an active site remodeled for catalysis of thioester bond formation[27] and that active site remodeling serves as a mechanism to drive the SUMO activation process forward. Interestingly, the ~180° rotation of the SCCH domain observed in the SUMO E1$^{COH000}$ structure resembles the SCCH domain closure that accompanies thioester bond formation; however, there is an additional ~50° rotation that results in the catalytic cysteine effectively swinging past the active site (Fig. 1a, b). As will be discussed in greater detail below, the conformation of the SCCH domain observed in the SUMO E1$^{COH000}$ structure is stabilized by interactions with the AAD

and IAD and results in the catalytic cysteine being positioned ~12 Å away from the active site. A final point worth noting is that because both the SUMO E1$^{APO}$ and SUMO E1$^{COH000}$ structures were determined from crystals belonging to the same space group (Table 1), crystal packing is unlikely to contribute to the network of SUMO E1 conformational changes observed upon COH000 binding (Supplementary Fig. 1 and 2).

**COH000 targets a cryptic binding pocket on SUMO E1.** Previous mass spectrometry and biochemical data indicate that SUMO E1 binding to COH000 is accompanied by formation of a covalent bond between Cys30 of the Uba2 subunit of SUMO E1 and an electrophilic center within the 7-oxabicyclohept-2-ene group of COH000 via Michael addition[41]. Uba2 Cys30 is located in the N terminal half of Helix 2 at a site within the SUMO E1 AAD that is proximal to the adenylation active site and, indeed, there is extensive electron density projecting from the Cys30 side chain into which all the atoms of COH000 could unambiguously be placed (Fig. 2a and Supplementary Fig. 3). Further, the covalent bond between Cys30 and COH000 is clear (Fig. 2a and Supplementary Fig. 3), confirming the previously reported mass spectrometry and biochemical data[41]. Importantly, there is no unaccounted for electron density in proximity to any of the other 17 cysteine residues of SUMO E1 (6 of which have solvent accessible surface areas >10 Å$^2$ in the open, closed, or COH00-bound states), a result that further supports the specificity of COH000 for Uba2 Cys30.

COH000 binds to a composite surface on SUMO E1 formed by amino acids from both the AAD and IAD (Fig. 2b) at a location distinct from the active site, and ~500 Å$^2$ from a total of ~600 Å$^2$ solvent accessible area of COH000 are buried upon complex formation. The *p*-methylbenzene group of COH000 inserts into a deep pocket on the surface of SUMO E1 where it engages in an extensive network of van der Waals interactions with Lys34, Val37, Leu38, Val80, Phe83, and Tyr84 on the Uba2 AAD, and Ala78 and Phe80 on the Sae1 IAD (Fig. 2b). The aniline group of COH000 engages in a network of van der Waals interactions with a hydrophobic patch near the entry of the pocket involving Phe83, and the aliphatic portion of Glu31 and Lys34 side chains of the Uba2 AAD, and Lys53, Phe80, and Leu102 of the Sae1 IAD (Fig. 2b). The only hydrogen bond observed at the SUMO E1/COH000 interface occurs between the nitrogen atom of the aniline group and the backbone carbonyl atom of Cys30 from the Uba2 AAD. In addition to the covalent bond to Cys30 of the AAD, the 7-oxabicyclohept-2-ene group of COH000 makes van der Waals contacts with Leu33, Ala76, and Ser79 of Uba2 AAD, and other than a few van der Waals contacts to Ser79 and Phe83 of the Uba2 AAD, the two methylester groups of COH000 are not observed to participate in extensive contacts with SUMO E1 (Fig. 2b).

The results of structure-guided biochemical experiments support the importance of the noncovalent interactions of COH000 for inhibition. Examination of a compound that is chemically related to COH000 and contains the same Michael acceptor but lacks the phenyl groups reveals that the phenyl groups are critical for SUMO E1 inhibition (Fig. 2c). Furthermore, disruption of the COH000 binding pocket on SUMO E1 via mutation of Phe80 in Sae1 and both Lys34 and Phe83 in Uba2 to alanines significantly diminishes the inhibitory effect of COH000 and, as expected, a Uba2 C30S mutant is not susceptible to inhibition by COH000[41] (Fig. 2d). It is worth noting that both of these SUMO E1 mutants exhibit reduced activity compared to wild type, further highlighting the sensitivity of SUMO E1 catalytic activity to structural perturbations in the COH000 binding region. Altogether, the data are consistent with the

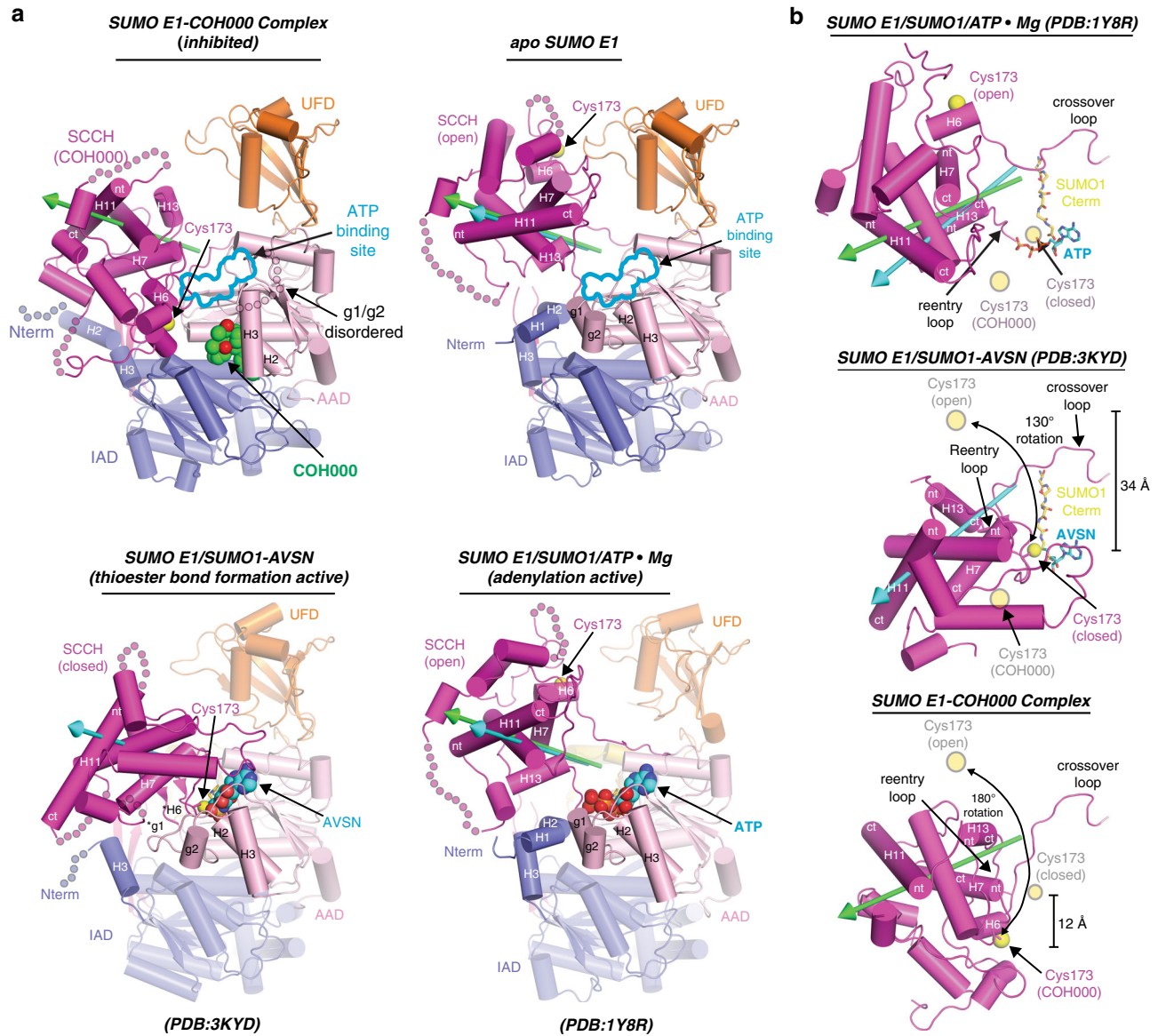

**Fig. 1** Overall structure of SUMO E1[COH000] and comparison to other SUMO E1 conformational snapshots. **a** Comparison of the SUMO E1[COH000], SUMO E1[APO], SUMO E1/SUMO1/ATP•Mg (PDB: 1Y8R), and SUMO E1/SUMO1-AVSN (PDB: 3KYD) structures, shown as cartoon representations. The green and cyan arrows highlight the rotation axes during transition of the SCCH domain from open to COH000-inhibited and open to closed conformations, respectively. AVSN is an adenosine analog harboring an electrophilic vinyl sulfonamide that was used to covalently trap the tetrahedral intermediate generated during SUMO E1–SUMO thioester bond formation[27,39]. Asterisks indicate helices that undergo structural remodeling in the thioester bond formation active structure. SUMO E1 domains are labeled and color-coded. COH000, ATP, AVSN, and the SUMO E1 catalytic cysteine are shown as spheres. Selected helices of the SCCH domain are labeled and their N and C termini are indicated by 'nt' and 'ct', respectively. The ATP-binding sites of the SUMO E1[COH000] and SUMO E1[APO] structures are indicated with a cyan outline. Regions of disorder are indicated with semitransparent circles. **b** Comparison of SUMO E1 SCCH domain in the open (adenylation active), closed (thioester bond formation active), and COH000 (inhibited) conformations. The adenylation domains (which serve as the rigid body of SUMO E1) were superimposed and the SCCH domains are shown as cartoons with the catalytic cysteines shown as spheres. SCCH domain alternations are highlighted by double-headed arrows and the degree of rotation between each conformational state (with reference to the open conformation) is indicated. Rotation axes of the SCCH domains are highlighted as in **a**. To provide a frame of reference, the relative position of the catalytic cysteine in the other SCCH domain conformational states are indicated with semitransparent yellow circles and labeled accordingly in each of the panels

importance of the extensive noncovalent interactions between COH000 and SUMO E1 and the specificity of COH000 for Uba2 Cys30 over the other 17 cysteine residues in SUMO E1.

Intriguingly, structural analysis reveals that the COH000 binding pocket observed in the SUMO E1[COH000] structure (including Cys30 of the Uba2 AAD) is completely buried by a stretch of residues spanning from Val54 to Val68 of the Uba2 AAD in all other structures of SUMO E1[26,27] (Fig. 2e).

Comparison of all previously determined adenylation active structural snapshots of SUMO E1 shows that this g1/g2 region of the AAD, named for its two 3₁₀ helices g1 and g2, is well ordered and adopts a similar conformation (Supplementary Fig. 4a). Although the g1/g2 region undergoes conformational changes during thioester bond formation[27], it is nevertheless well ordered and similarly occludes the COH000 binding site (Fig. 2e). In contrast, analysis of the SUMO E1[COH000] structure reveals a

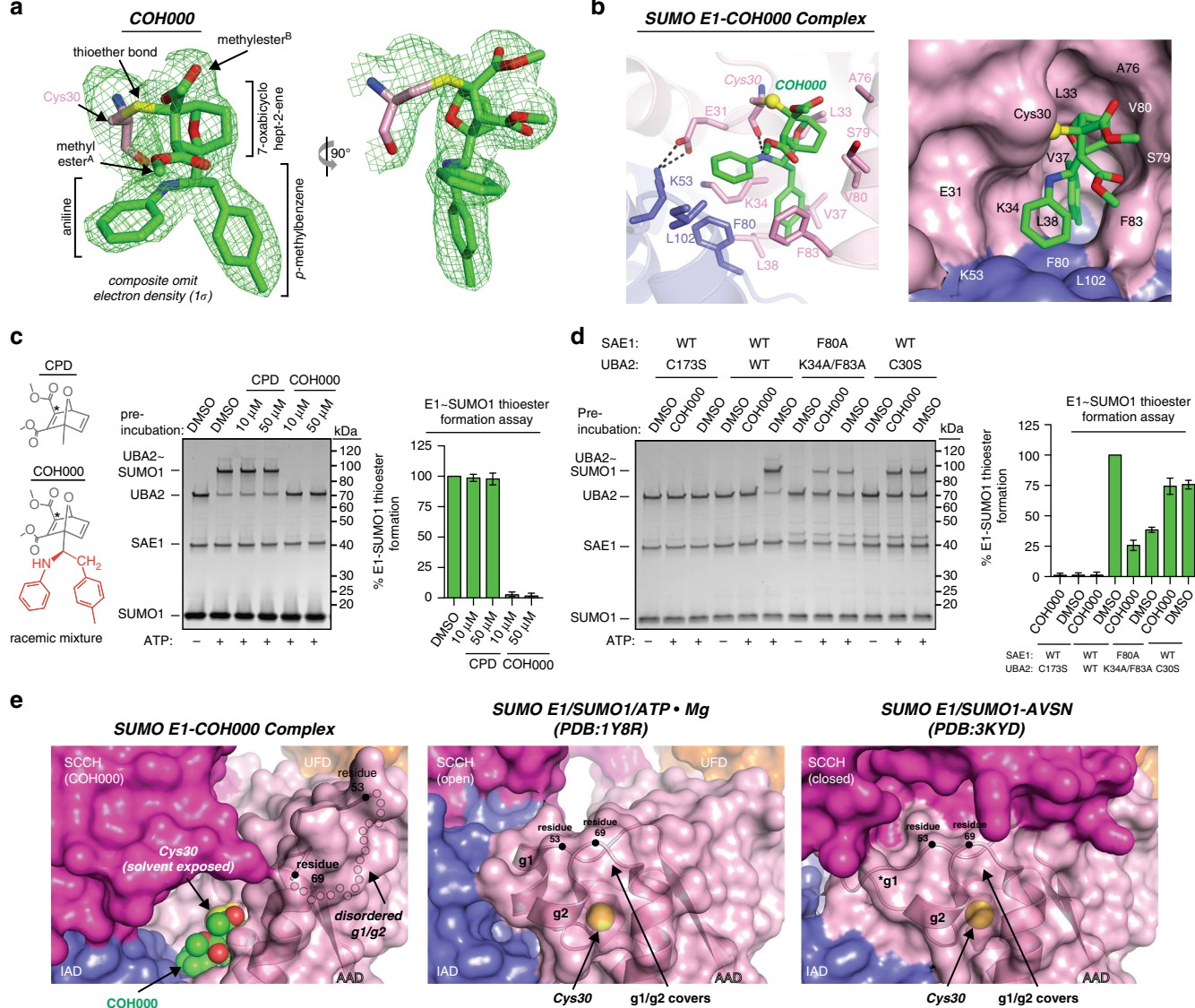

**Fig. 2** COH000 targets a cryptic binding site on SUMO E1. **a** Composite omit electron density map (contoured at 1σ) for COH000 is shown as green mesh. COH000 is shown as sticks with carbons (green), oxygens (red), and nitrogens (blue). Cys30 of the AAD is shown as sticks with sulfur (yellow). **b** (top) The SUMO E1$^{COH000}$ structure is shown as a cartoon (colored as in Fig. 1a), with COH000-interacting residues shown as sticks. Hydrogen bonds are shown as dashed lines. **b** (bottom) SUMO E1 is shown as a surface representation to highlight the COH000 binding pocket. SUMO E1 residues involved in contacts to COH000 are labeled. **c** The core structure containing the Michael acceptor of COH000 is not sufficient to achieve inhibition. The compound L163392 from Aldrich (labeled CPD) was compared side by side with COH000 for biochemical inhibition of SUMO E1 using SUMO E1-SUMO1 thioester formation assays as described in the Methods. Data from three independent experiments were quantitated and presented as bar graphs, with error bars representing ± 1 SD. Chemical structures are shown on the left with the Michael acceptor of COH000 highlighted with a black star. **d** SUMO E1–SUMO1 thioester formation assays were performed with the indicated SUMO E1 variants in the presence and absence of COH000 as described in the Methods. Data were quantitated as in **c**. C173S serves as a catalytically inactive negative control. **e** The indicated SUMO E1 structures are shown as semitransparent surface representations with selected regions shown as cartoons. COH000 is shown as spheres. The disordered g1/g2 region in the SUMO E1$^{COH000}$ structure is shown as semitransparent circles and the residues marking the beginning and end of this disordered region (residues 53 and 69) are highlighted with a black circle in all the structures to allow for direct comparison. The asterisk highlights the g1 helix that undergoes structural remodeling in the thioester bond formation active structure

complete disordering of the g1/g2 region that exposes both Cys30 and the COH000 binding pocket to solvent (Fig. 2e). In the context of previous studies demonstrating a transition between open and closed SUMO E1 states that occurs during catalysis of adenylation and thioester bond formation, this observation suggests that COH000 gains access to its cryptic binding pocket either as a result of the g1/g2 region adopting a previously unobserved conformation during this transition, or that the g1/g2 region exhibits a much greater degree of conformational flexibility than suggested by previous crystal structures. As will be discussed in detail below, binding of COH000 to this site and concomitant disordering of the g1/g2 region is significant because this region harbors several residues important for catalysis and its disordering is associated with a series of additional conformational changes in SUMO E1 that altogether account for the COH000 mechanism of inhibition.

**Structural basis for inhibition of SUMO E1 by COH000.** Analysis of the SUMO E1$^{COH000}$ structure reveals that the COH000 mechanism of inhibition involves induction of both local (in the AAD and IAD) and broader large-scale conformational changes (in the SCCH domain) that effectively disassemble both the adenylation and thioester bond formation active sites (Supplementary Movie 1). Docking of COH000 onto previously determined structures shows that COH000 clashes with the g1/g2 region in both the adenylation and thioester bond formation active conformations of SUMO E1 (Fig. 3a and Supplementary Fig. 4b). This suggests that once COH000 gains access to the binding pocket and forms a covalent adduct with Cys30, steric occlusion prevents the g1/g2 helix from adopting the conformations observed in previous structures, thereby contributing to its disordering. This is important because the g1/g2 region harbors several residues critical for adenylation, including Asp48, Leu49, Ser55, Asn56, Arg59, and Lys72, which make contacts to ATP (Fig. 3b), and Asp50 which engages in hydrogen bonds to the backbone amides of Asn177 and Thr178 of the Uba2 SCCH domain that are crucial for thioester bond formation[26,27] (Supplementary Fig. 4c). The g1/g2 region (predominantly the g1 helix) also indirectly contributes to adenylation by engaging in contacts that stabilize the N-terminal region of the Sae1 IAD, around α-helices 1 and 2 (H1$^{IAD}$ and H2$^{IAD}$), resulting in proper positioning of IAD residues important for catalysis, including Arg21, which contacts the γ-phosphate of ATP[26,27] (Fig. 3b). Residues 1 to 16 of Sae1, which includes all of H1$^{IAD}$, are disordered in the SUMO E1$^{COH000}$ structure, and although H2$^{IAD}$ is ordered, a significant rotation about a hinge located in the short linker between H2 and H3 of the IAD effectively swings H2$^{IAD}$ away from the active site, displacing residues such as Sae1 Arg21 that are important for adenylation (Fig. 3b). We surmise that this conformational change in the IAD results from the loss of contacts with the g1/g2 region due to its disordering upon COH000 binding.

An additional conformational change observed upon COH000 binding occurs around the N terminus of H2 in the Uba2 AAD (H2$^{AAD}$), which comprises part of the ATP-binding pocket of SUMO E1 and harbors the oxyanion hole of the active site. H2$^{AAD}$ plays a critical role in catalysis by engaging in hydrogen bonds (Gly27 and Ile28 backbone amines) to the α-phosphate of ATP and the carbonyl oxygen of Gly97 of SUMO1 and providing positive electrostatic potential (through the helix dipole) that together stabilize the transition states and intermediates formed during adenylation and thioester bond formation[27] (Fig. 3c). As expected, given its key role in catalysis, H2$^{AAD}$ adopts nearly the same conformation in all previous structures of SUMO E1[26,27], with the carbonyl oxygens of Gly26 and Gly27 participating in the α-helical hydrogen bonding network and representing the start of H2 (Supplementary Fig. 4a). In contrast, binding of COH000 to SUMO E1 and covalent adduct formation to Cys30 results in a distortion of the structure of the N terminus of H2, including the oxyanion hole (Fig. 3c). Specifically, a significant alteration in the phi and psi angles of Gly26 and Gly27 of the AAD disrupts the α-helical hydrogen bonding network and shortens H2 by one turn of the helix such that H2$^{AAD}$ now begins at Gly29 (Fig. 3c). Though subtle, this conformational change would result in steric clashes with both the α-phosphate of ATP and the carbonyl oxygen of SUMO1 Gly97 and alters the capacity of this region to engage in the hydrogen bonding and electrostatic interactions that are crucial for catalysis based on all previous structures of E1 enzymes in complex with ATP and adenylate intermediates. An additional local conformational change observed in the SUMO E1$^{COH000}$ structure is an ~2 Å translation of H3$^{AAD}$ that presumably occurs due to steric clashes with COH000 (Fig. 3c). Though a potential mechanistic consequence of the H3$^{AAD}$ shift is unclear, it may contribute to disordering of the g1/g2 region which begins immediately after H3$^{AAD}$.

**COH000 exploits conformational coupling of SUMO E1 domains.** Conformational coupling of the adenylation and SCCH domains plays an important role in the catalytic mechanism of SUMO E1 through its role in remodeling the active site and thereby toggling its ability to catalyze adenylation and thioester bond formation[26,27]. While the conformational status of the SUMO E1 g1/g2$^{AAD}$ region is connected to the conformation of the H1/H2$^{IAD}$ region as noted above, the H1/H2$^{IAD}$ region is in turn connected to the conformation of the SCCH domain. In adenylation active snapshots of SUMO E1, the SCCH domain is perched in the open conformation on a platform comprising H1$^{IAD}$ and H2$^{IAD}$ [26,27] (Fig. 4a). In the thioester bond formation active conformation, H1$^{IAD}$ and H2$^{IAD}$ become completely disordered, disrupting IAD contacts with the SCCH domain so that the SCCH domain is free to undergo the ~130° rotation that positions the active site cysteine (Cys173) proximal to the SUMO C terminus where catalysis takes place[27] (Fig. 4a). H1/H2$^{IAD}$ disordering also creates the space required for the SCCH domain to adopt the closed conformation which is stabilized by a network of contacts with AAD that effectively covers the active site during thioester bond formation[27] (Fig. 4a).

In the SUMO E1$^{COH000}$ structure, H1$^{IAD}$ is disordered and H2$^{IAD}$ undergoes a conformational change that swings it out of the active site. This again disrupts contacts between the IAD and SCCH domain, freeing the SCCH domain to undergo an ~180° rotation (Supplementary Movie 1) that resembles the ~130° rotation that accompanies thioester bond formation (Fig. 1). The additional 50° rotation of the SCCH domain that accompanies COH000 binding results in a completely different network of contacts at the SCCH/AAD and SCCH/IAD (particularly H2$^{IAD}$) interfaces compared to adenylation and thioester bond formation active snapshots of SUMO E1 (Fig. 4a). Intriguingly, when the SUMO E1$^{COH000}$ SCCH domain is modeled onto adenylation active SUMO E1, H6 and the H7-g3 loop of the SCCH domain directly overlap with g2 of the AAD and H1/H2$^{IAD}$ (Fig. 4b). Thus, stabilization of the SCCH domain in the conformation observed in the SUMO E1$^{COH000}$ structure sterically blocks reassembly of the adenylation active site, providing another layer to the mechanism by which COH000 inhibits SUMO E1. Altogether, analysis of the structures reveals that the COH000 mechanism of inhibition involves allosteric coupling of the adenylation and SCCH domains that is enabled through a remarkable exploitation of SUMO E1 conformational plasticity that is required for its normal catalytic activity.

Finally, the RLW motif located on H2 of the Sae1 IAD (Arg24, Leu25, and Trp26) plays an important role in stabilization of the SCCH domain in its distinct adenylation active[26,27], thioester bond formation active[27], and COH000-inhibited conformations. Specifically, differences in RLW motif-mediated networks of intra- and interdomain contacts stabilize different H1/H2$^{IAD}$ conformational states and result in distinct IAD surfaces that differentially engage and stabilize the SCCH domain in its adenylation active and COH000-inhibited conformations (Fig. 4c). While Arg24 of the IAD engages in contacts to Asp20 of the IAD, and Phe374 and Ile383 of the SCCH domain in adenylation active snapshots of SUMO E1[26,27], this residue makes no contacts in the SUMO E1$^{COH000}$ structure because of the COH000-induced H2$^{IAD}$ conformational change (Fig. 4c). Similarly, Leu25 of the IAD, which engages in van der Waals contacts to Tyr144 of the AAD and Pro385 and Ile387 of the SCCH domain in adenylation active SUMO E1[26,27], participates in no interactions in the SUMO E1$^{COH000}$ structure (Fig. 4c).

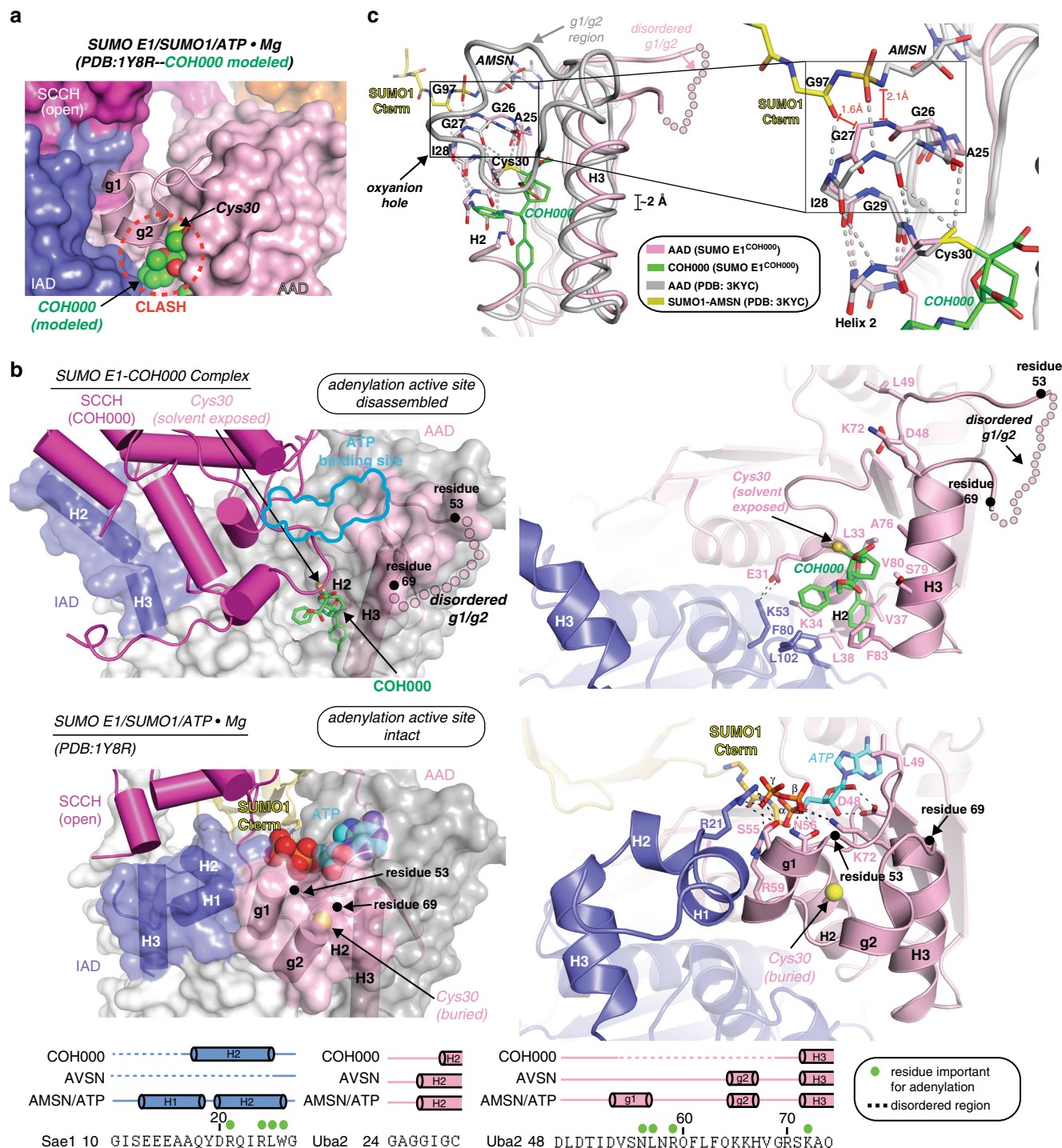

**Fig. 3** Small- and large-scale conformational changes are induced in SUMO E1 upon COH000 binding. **a** COH000 (spheres) was docked onto SUMO E1 from the SUMO E1/SUMO1/ATP•Mg structure (PDB: 1Y8R). SUMO E1 is shown as a surface representation with the exception of the g1/g2 region, which is shown as cartoon. The steric clash between COH000 and the g1/g2 region is highlighted by a dashed red circle. Two views of the structure, rotated by 90° about the y-axis are presented. **b** The SUMO E1[COH000] (top) and SUMO E1/SUMO1/ATP•Mg (PDB: 1Y8R; middle) structures are shown in the same orientation to highlight conformational changes that occur upon COH000 binding. Regions undergoing conformational changes are colored by their domain as in Fig. 1a and the rest of the structure is colored light gray. For clarity, the SCCH domains are not shown in the magnified views presented in the right panels. **b** (bottom) Structure-based sequence alignment of SUMO E1 active site regions undergoing conformational changes upon COH000 binding. Secondary structure for the indicated structures are shown above the sequence colored with dashed lines indicating disorder. Residues important for catalysis of adenylation are highlighted by green circles. **c** (left) The adenylation domains of the SUMO E1[COH000] and SUMO E1/SUMO1-AMSN (PDB: 3KYC) structures were superimposed and are shown as colored light pink and gray tubes, respectively. AMSN is an adenosine analog harboring a sulfamide group that was used to generate a nonhydrolyzable mimic of the SUMO1-adenylate intermediate[27,39]. The N-terminal half of AAD H2 is shown as sticks and the oxyanion hole is boxed. The 2 Å shift of H3 induced upon COH000 binding is indicated and the disordered g1/g2 region of the SUMO E1[COH000] structure is shown as semitransparent light pink circles. The C terminus of SUMO1 (yellow) and AMSN (gray) are shown as sticks. **c** (right) Magnified view around the SUMO E1 oxyanion hole

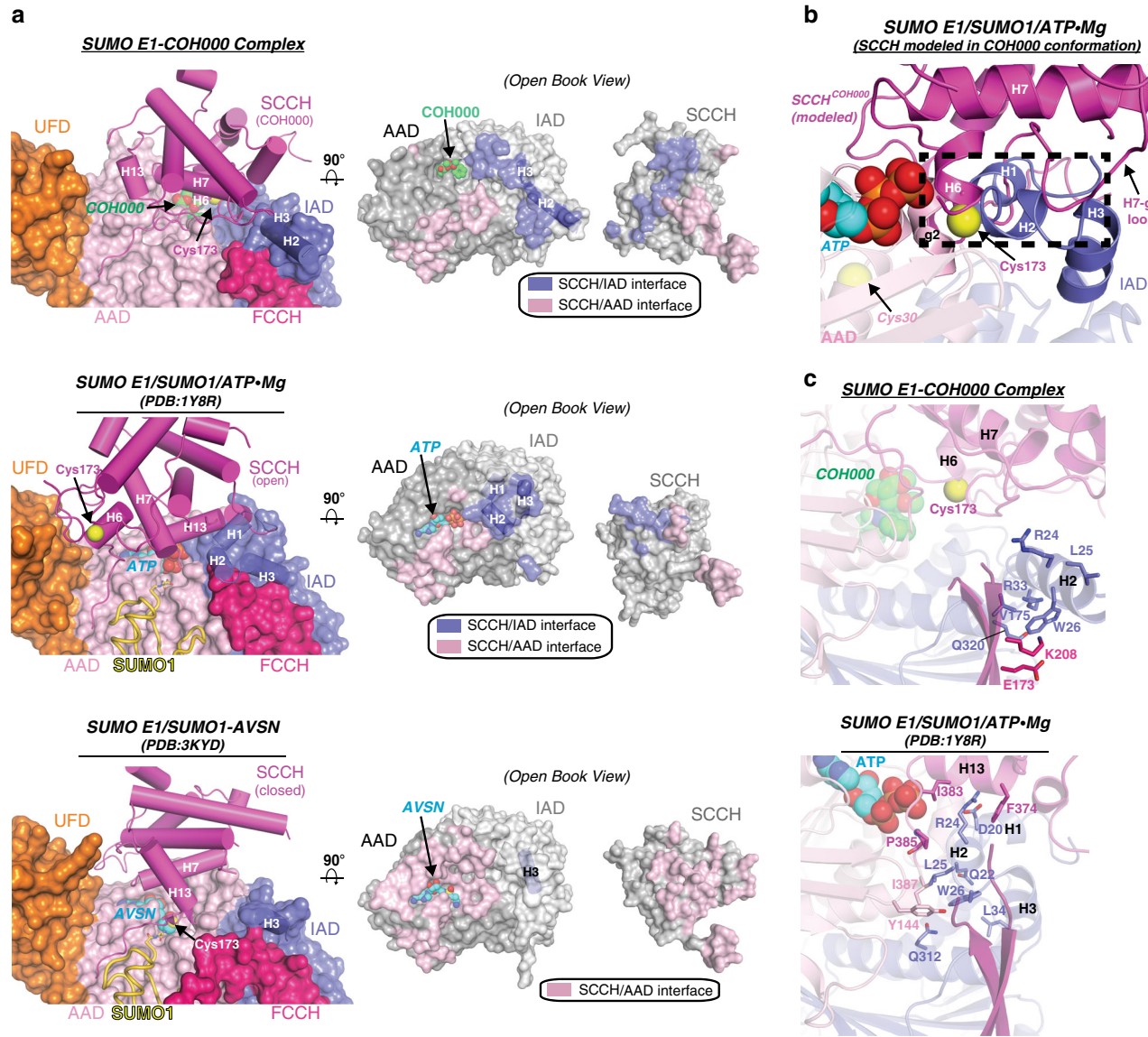

**Fig. 4** COH000 exploits allosteric coupling of the SUMO E1 adenylation and SCCH domains. **a** (left) The adenylation domains and UFD of the indicated structures are shown as surface representations and the SCCH domains are shown as cartoons. The SUMO E1 catalytic cysteines, COH000, ATP, and AVSN are shown as spheres. **a** (right) Open book representation of the indicated structures shown in a top-down view relative to the left panel. Residues buried at the IAD/SCCH and AAD/SCCH interfaces are colored slate and pink, respectively. **b** The SCCH domain is modeled onto an adenylation active SUMO E1$^{COH000}$ structure in the COH000-inhibited conformation. The model was created by superimposing the adenylation domains of the SUMO E1$^{COH000}$ and SUMO E1/SUMO1/ATP•Mg (PDB: 1Y8R) structures. The SUMO E1/SUMO1/ATP•Mg structure is shown as cartoon, and the SCCH domain is not shown for clarity. Only the SCCH domain from the SUMO E1$^{COH000}$ structure is presented (as a cartoon). The SUMO E1 catalytic cysteine (Cys173), Cys30, and ATP are shown as spheres. Regions involved in steric clashes between the adenylation domains and the SCCH domain modeled in the COH000 conformation are highlighted with a dashed black box. **c** The indicated structures are shown as cartoon representations. The RLW motif of the Sae1 IAD and SUMO E1 residues involved in contacts to the RLW motif are shown as sticks

Interestingly, while Trp26 interacts with Tyr144 of the AAD, Ile387 of the SCCH domain, and Leu34 and Gln312 of the IAD in adenylation active structures[26,27], this residue engages in a completely different network of contacts in the SUMO E1$^{COH000}$ structure, including Arg33, Glu173, Val175, Lys208, and Gln320 of the IAD (Fig. 4c). The Trp26 side chain is well ordered in the SUMO E1$^{COH000}$ structure and the unique set of interactions resulting from the COH000-induced H2$^{IAD}$ rotation likely contributes to stabilization of the active site in an inactive conformation. Therefore, the conformational variability of SUMO E1 architecture is intimately linked to the plasticity of intra- and interdomain amino acid interaction networks that occur within the enzyme.

**Comparison of the COH000 binding pocket in other Ubl E1s.** There are eight known Ubl E1 enzymes that all activate their cognate Ubls via a conserved two-step mechanism involving adenylation followed by formation of a thioester bond between the E1 catalytic cysteine and the Ubl C terminus[2,3]. SUMO E1, along with the E1s for Ub (Uba1 and Uba6), Nedd8 (AppBP1/Uba3), and Isg15 (Uba7), are classified as canonical E1s due to their similar overall domain organization and structure, while the more divergent E1s for Urm1 (Uba4), Ufm1 (Uba5), and Atg8/Atg12 (Atg7) are classified as noncanonical[42]. The adenylation domains of all Ubl E1s exhibit a high degree of sequence and structural similarity, and analysis reveals potential cryptic pockets at regions corresponding to the COH000 binding site of SUMO

E1 that are completely covered by the g1/g2 region of the respective Ubl E1s (Supplementary Fig. 5). To assess the specificity of COH000 for SUMO E1 versus other Ubl E1s, a panel of Ubl E1–Ubl thioester formation assays was performed under equivalent conditions. The results demonstrate that COH000 strongly inhibits SUMO E1, as expected, but exhibits little ability to inhibit other canonical Ubl E1s, including Uba1, Uba6, Nedd8 E1 or the noncanonical Ubl E1, Uba5 (Fig. 5a). This observation is consistent with analysis of an overlapping but distinct subset of Ubl E1s[41]. In order to gain insights into the

molecular basis for the specificity of COH000 for SUMO E1, we first performed a structure-based sequence alignment in the regions of the AAD and IAD harboring residues in direct contact with COH000 in the SUMO E1[COH000] structure. The results of this analysis show that there is a significant degree of variability in the physicochemical properties of COH000-interacting residues in other Ubl E1s, with amino acid identity ranging from 14 to 71% and similarity ranging from 57 to 93% (Fig. 5b).

As expected for noncanonical Ubl E1s (Uba4, Uba5, and Atg7) that share limited identity/similarity with SUMO E1 in this

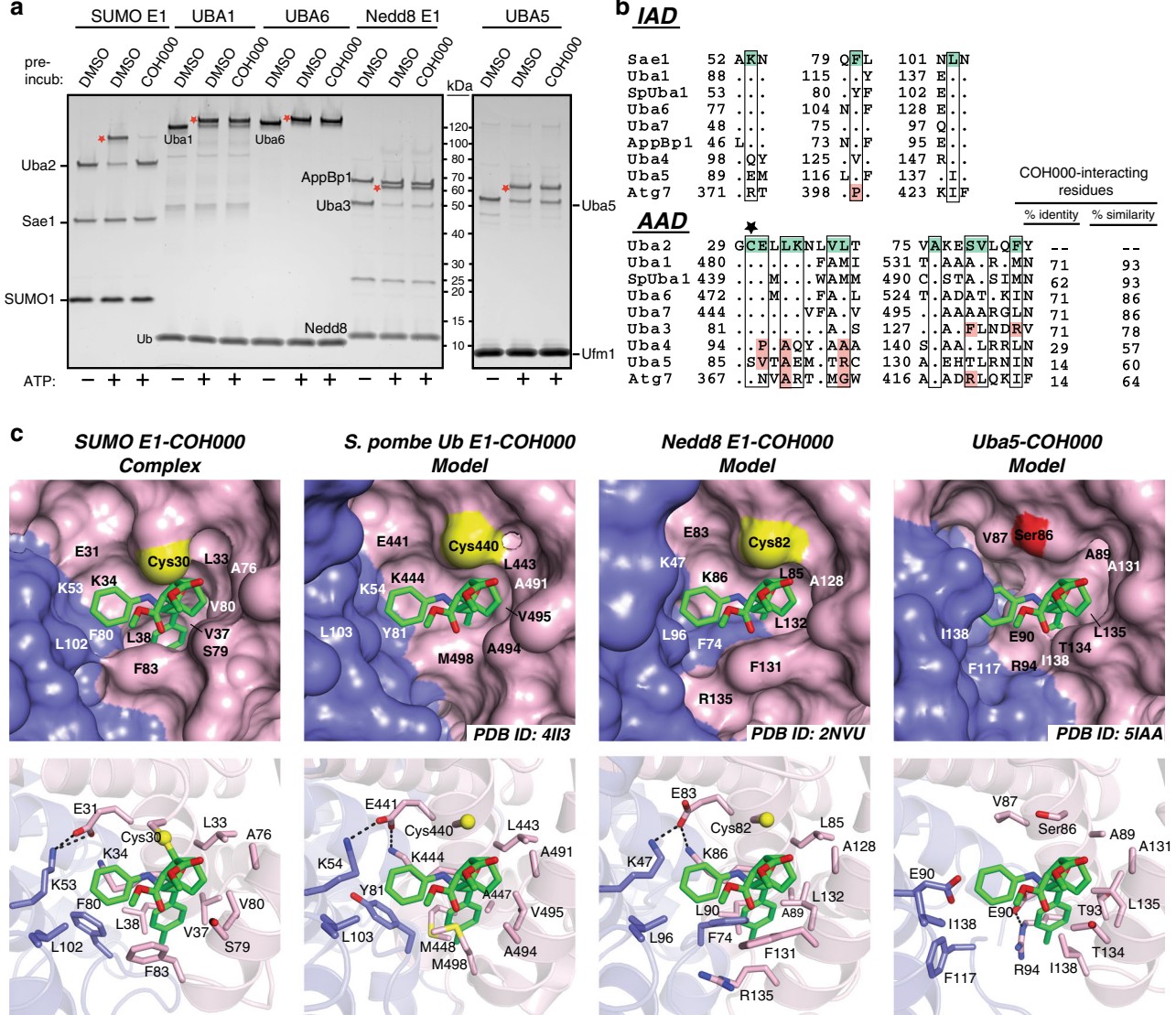

**Fig. 5** Insights into the molecular basis of COH000 specificity for SUMO E1. **a** In vitro E1–Ubl thioester formation assays were performed as described in the Methods. The samples were subjected to SDS-PAGE and visualized with Sypro Ruby stain. The E1-Ub thioester products are indicated with a red star. **b** Structure-based sequence alignment of the COH000-interacting region of the indicated Ub and Ubl E1 enzymes. Residues involved in contacts to COH000 in the SUMO E1[COH000] structure are shaded green and the cysteine residue that forms the thioether bond with COH000 (Cys30) is indicated with a black star. Residues of other Ubl E1s at positions corresponding to the COH000 binding site of SUMOE E1 that are identical are indicated by periods and those that harbor different physicochemical properties are shaded red. The percent identity and similarity of Ubl E1 residues corresponding to the COH000 binding site of SUMO E1 are shown to the right of the alignment. 'Sp' stands for *S. pombe*. **c** Models of Ubl E1-COH000 complexes were created by superimposing the adenylation domains of Ubl E1s onto the SUMO E1[COH000] structure. **c** (top) The indicated Ubl E1 structures are shown as surface representations with a magnified view of the region corresponding to the COH000-interacting pocket observed in the SUMO E1[COH000] structure. The g1/g2 regions of the Ubl E1s are not shown because they occlude the cryptic COH000 binding pocket. The SUMO E1[COH000] structure is shown in the same orientation in the left panel for comparison. Cys30 of SUMO E1 and the corresponding cysteines/serines of other Ubl E1s are colored yellow and red, respectively. Ubl E1 residues corresponding to the COH000 binding site of SUMO E1 are labeled. **c** (bottom) The structures are shown as cartoon representations with Ubl E1 residues corresponding to the COH000 binding site of SUMO E1 shown as sticks and labeled

region, analysis of the enzyme surface in the region corresponding to the COH000 binding site of SUMO E1 reveals pockets with significant differences in shape when the g1/g2 regions of the E1s are removed (Fig. 5c). In contrast to canonical Ubl E1s that are either single polypeptides (Uba1, Uba6, and Uba7) or obligate heterodimers (SUMO and Nedd8 E1s) that harbor one functional active site, Uba4, Uba5, and Atg7 each function as homodimers in solution with two functional active sites. As a result, a major fraction of the predicted COH000 binding pocket of one protomer of homodimeric E1s is formed by the g1/g2 region of the other protomer in the homodimers (Supplementary Fig. 6). Since the g1/g2 region must be displaced in order for COH000 to access its cryptic binding site in SUMO E1, this suggests that only one of the two protomers in Uba4, Uba5, and Atg7 could be targeted by an COH000-like molecule, since binding of the molecule to one protomer disrupts the binding site of the other.

Canonical Ubl E1s each harbor 71% sequence identity at positions observed to interact with COH000 in the SUMO E1[COH000] structure and sequence similarities ranging from 78 to 93% (Fig. 5b). Despite these very high degrees of sequence similarity, analysis of the crystal structures of these E1s reveals that the shape of COH000 binding pockets in these E1s varies significantly (Fig. 5c). This variability is largely the result of plasticity in the intramolecular interactions that take place at and around the COH000 binding site even for amino acids that are identical in the different structures. For example, Tyr81 (Phe116, human numbering) in the IAD of *Schizosaccharomyces pombe* Uba1 (which corresponds to Phe80 of the SUMO E1 IAD) is involved in intradomain contacts, whereas the corresponding residue of Nedd8 E1 (Phe74 of the AppBP1 subunit) is involved in interdomain contacts to residues in the AAD (Fig. 5c). Thus, while the amino acid identities and similarities of residues in the COH000 binding site of other E1s are very high, this region of the structure exhibits relatively subtle structural differences that alter the properties of the binding pocket. This suggests that a complex combination of sequence and structural variability across Ubl E1s likely accounts for the high degree of SUMO E1 specificity exhibited by COH000. It is worth noting, however, that the general conservation of this pocket in other Ubl E1s, together with the molecular basis by which COH000 inhibits SUMO E1 and the conserved mechanism for E1 activities, raises the possibility that this site could potentially be targeted by molecules related to COH000.

## Discussion

The structure of SUMO E1 in complex with COH000 presented in this manuscript reveals a completely unexpected molecular mechanism of inhibition. COH000 interaction with its cryptic SUMO E1 binding pocket and covalent adduct formation with AAD Cys30 are accompanied by a remarkable network of conformational changes that include distortion of the E1 oxyanion hole, disordering and conformational changes in the g1/g2[AAD] and H1/H2[IAD] regions that are crucially involved in catalysis, and a ~180° rotation of the catalytic cysteine-containing SCCH domain. Notably, COH000 binding to SUMO E1 is accompanied by local unfolding and conformational changes that couple the adenylation and SCCH domains. Altogether, these conformational changes disassemble the SUMO E1 adenylation and thioester bond formation active sites and result in new networks of intramolecular contacts that likely help to lock the enzyme in an inactive conformation.

Interestingly, COH000 exploits salient mechanistic features of the catalytic cycle of SUMO E1; specifically, disassembly of the adenylation active site, and SCCH domain alternation, which

normally serves to reposition the catalytic cysteine proximal to the SUMO C terminus where thioester bond formation takes place. The primary difference between the conformational changes accompanying thioester bond formation by SUMO E1 and COH000-mediated inhibition is that while the g1/g2 region is remodeled to form a critical part of the thioester bond formation active site, COH000 binding results in complete disordering of the g1/g2 region, thus preventing it from adopting both the adenylation and thioester bond formation-competent conformations. The SCCH domain also rotates by an additional 50° relative to the closure that accompanies thioester bond formation, adopting a conformation that sterically blocks reassembly of the active site.

An additional significant finding of this study is that the structure of SUMO E1 in the apo form (i.e., without ATP or SUMO) determined from crystals grown in the same space group and crystal packing environment as the SUMO E1[COH000] structure adopts the adenylation active conformation with the SCCH domain in the open conformation. This highlights that active site remodeling and SCCH domain alternation are indeed intrinsic and interconnected structural features of Ubl E1s[27,40] and that transitions between the adenylation and thioester bond formation active forms of E1 do not require Ubl or ATP binding, or ATP hydrolysis that occurs during adenylation. Rather, these SUMO E1 conformational states are in an equilibrium that is shifted towards one direction or the other depending on the nature of the substrate bound at the active site.

The cryptic nature of the COH000 binding pocket and the network of conformational changes in SUMO E1 that accompany binding and covalent adduct formation between COH000 and AAD Cys30 are unexpected and would be very difficult to identify or predict using current in silico methods. Thus, this study highlights the value of traditional drug development efforts combining high-throughput screening and structural studies to determine mechanism. Finally, the conservation of cryptic pockets in other Ubl E1s at the region corresponding to the COH000 binding site of SUMO E1 suggests that our SUMO E1[COH000] structure may provide a framework for the development of small molecules targeting the E1 enzymes for other Ubl pathways which are also targets for therapeutic intervention in human pathologies.

## Methods

**Protein expression and purification.** Human *SAE1* (residues 1–349) was cloned into vector *pET11c*. Human *UBA2* (residues 1–640) and human *SUMO1* (residues 1–97) were cloned into vector *pET28b* with a thrombin-cleavable N-terminal 6× His tag. Human *UBA1*[ΔNT] (residues 49–1058) was cloned into vector *pSMT3* to introduce an N-terminal Ulp1-cleavable 6× His-SMT3 tag[43,44]. Human *UBA6*[ΔNT] (residues 37–1052) was cloned into vector *pFastBac HTB* with an N-terminal TEV-cleavable 6× His tag. Human *APPBP1* (residues 1–534) and human *UBA3* (residues 22–463) were amplified from human complementary DNA (cDNA) library and were inserted into vector *pGEX6p-2* and *pET28*, respectively. Human *UBA5* (residues 1–404) was amplified from human cDNA library into vector *pSMT3*. Human *Ub* (residues 1–76) and human *Ufm1* (residues 1–83) were both amplified from human cDNA library into vector *pET29* with an N-terminal TEV-cleavable 6× His tag. Human *NEDD8* (residues 1–76) was amplified from a cDNA library into vector *pET28b*.

All proteins were recombinantly expressed in *Escherichia coli* BL21 (DE3) Codon Plus strain except for UBA6. Heterodimers of SUMO E1 (hSAE1/hUBA2) and Nedd8 E1 (APPBP1/UBA3) were expressed by co-transforming plasmids into BL21 (DE3) *E. coli*. Protein expression was induced with 0.5 mM isopropyl-β-D-1-thiogalactoside (IPTG) overnight at 18° C at OD$_{600}$ 1.0. Recombinant baculoviruses for UBA6 expression were generated using the Bac-to-Bac Baculovirus Expression System according to the manufacturer's instructions (Thermo Fisher Scientific).

Bacterial and insect cell pellets were harvested, resuspended, and lysed by sonication in lysis buffer (20 mM Tris HCl pH 8.0, 350 mM NaCl, 20 mM Imidazole, 2 mM 2-Mercaptoethanol (βME)). Supernatant was separated by centrifugation, and the supernatant applied to Ni-NTA agarose (Qiagen). All affinity tags were cleaved with Ulp protease, TEV protease, or thrombin, and further purified by gel filtration (Superdex 200 and Superdex 75; GE Healthcare), and anion-exchange (MonoQ 10/100 and MonoS 10/100; GE Healthcare) chromatography.

**SUMO E1$^{APO}$ and SUMO E1$^{COH000}$ complex crystallization**. SUMO E1$^{COH000}$ complex was formed by mixing SUMO E1 (25 mg/ml) with a twofold molar excess of COH000 and incubating for 15 min at 4° C. Initial crystallization of SUMO E1$^{APO}$ and SUMO E1$^{COH000}$ was performed in sitting drop 96-well format by mixing 0.2 µl of SUMO E1$^{APO}$ or SUMO E1$^{COH000}$ with 0.2 µl reservoir solution (0.2 M ammonium sulfate, 0.1 M Bis-Tris HCl pH 6.5, 20% PEG3350) at 18° C. Poorly diffracting crystals appeared after 2 months and were used to prepare seed stocks. Crystals used for data collection and structure determination were obtained after two rounds of microseeding performed using hanging drop vapor diffusion in 24-well plates at 15° C by mixing 1 µl SUMO E1 (18 mg/ml), 1 µl reservoir solution (0.2 M ammonium sulfate, 0.1 M Bis-Tris HCl pH 6.5, 20% PEG3350), and 2 µl freshly prepared seed stock (1:100,000 dilution). Resulting crystals were cryoprotected in reservoir solution supplemented with an additional 10% PEG3350 and 10% glycerol, and subsequently flash frozen in liquid nitrogen.

**Structure determination and refinement**. A complete data set was collected from the SUMO E1$^{APO}$ crystals to 3.1 Å resolution at the Advanced Photon Source (APS), SER-CAT beamline 22-ID at a wavelength of 1.08 Å. A complete data set was collected from the SUMO E1$^{COH000}$ crystals to 2.45 Å resolution at APS, NE-CAT beamline 24-IDE at a wavelength of 1.00 Å. All data were indexed, integrated, and scaled using HKL2000[45]. SUMO E1$^{APO}$ and SUMO E1$^{COH000}$ crystals both belong to space group P2$_1$2$_1$2$_1$ with unit cell dimensions $a = 56.1$, $b = 115.4$, $c = 173.0$ and $a = 56.1$, $b = 116.0$, $c = 174.1$, respectively. Both crystals have one copy of SUMO E1 per asymmetric unit.

The structures were solved by molecular replacement using the program PHASER[46]. The search model used for the SUMO E1$^{APO}$ structure was SUMO E1 from the SUMO E1/SUMO E1-AMSN adenylate intermediate structure[27] (PDB: 3KYC) and the search model used for the SUMO E1$^{COH000}$ structure contained the same coordinates with the SCCH domain deleted. After one round of refinement, the resulting maps were inspected and the SCCH domain was placed into unambiguous electron density for the SUMO E1$^{COH000}$ structure. Regions of the model that did not fit the electron density due to conformational changes were subsequently removed and the models were subjected to iterative rounds of refinement and rebuilding using PHENIX[47] and COOT[48]. COH000 was built into its corresponding electron density in the SUMO E1$^{COH000}$ structure during the final rounds of refinement. The final models have R/Rfree values of 0.225/0.258 (SUMO E1$^{APO}$) and 0.193/0.236 (SUMO E1$^{COH000}$) and have excellent geometry as assessed using Molprobity[49]. Ramachandran plot statistics for the SUMO E1$^{APO}$ structure: favored (94.2%), allowed (5.7%), and outliers (0.1%). Ramachandran plot statistics for the SUMO E1$^{COH000}$ structure: favored (96.1%), allowed (3.8%), and outliers (0.1%).

**E1 inhibition assays**. All inhibition assays were performed by incubating 0.25 µM of the indicated E1 with 2 µM COH000 (or CPD) in a buffer containing 20 mM HEPES-Na, pH 7.5, 50 mM NaCl, 5 mM MgCl$_2$, and 5% dimethyl sulfoxide for 10 min at room temperature. ATP and the corresponding Ubls were subsequently added to a final concentration of 1 mM and 4 µM respectively. Reactions were incubated at room temperature for 20 min and terminated using non-reducing urea SDS loading buffer. Samples were resolved on 4–15% gradient gel (Biorad) at 180 V and visualized with SYPRO Ruby stain (Biorad). Image was taken using Geldoc (Biorad), and processed with Imagelab (Biorad). All assays were performed in triplicate and the results of a representative experiment are presented.

**Chemical synthesis of COH000**. Synthetic procedures and characterization of COH000 are described in the Supplementary Note. See also Li et al.[41].

**Reporting summary**. Further information on research design is available in the Nature Research Reporting Summary linked to this article.

## Data availability

Atomic coordinates and structure factors are deposited in the RCSB with accession codes 6CWY and 6CWZ. The data that support the findings of this study are available from the corresponding authors upon reasonable request.

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

## Acknowledgements
The authors thank Miklos Bekes and Thomas D.Y. Chung for critically reading the manuscript. X-ray diffraction data were collected at SER-CAT 22-ID and NE-CAT 24-ID-E beamlines at the Advanced Photon Source, Argonne National Laboratory. This work is based upon research conducted at the Northeastern Collaborative Access Team beamlines, which are funded by the National Institute of General Medical Sciences from the National Institutes of Health (P41 GM103403). The Pilatus 6M detector on 24-ID-C beamline is funded by a NIH-ORIP HEI grant (S10 RR029205). This research used resources of the Advanced Photon Source, a U.S. Department of Energy (DOE) Office of Science User Facility operated for the DOE Office of Science by Argonne National Laboratory under Contract No. DE-AC02-06CH11357. The X-ray crystallography facility used for this work is supported by the Office of the Vice President for Research at the Medical University of South Carolina. The liquid handling robot used was purchased via an NIH Shared Instrumentation Award (S10 RR027139-01). Research reported in this publication was supported by the NIH R01 GM115568 (to S.K.O.), R01GM086171, R01GM102538, and R03DA026556 (to Y.C.). This work was also supported, in part, by the Hollings Cancer Center's Ruth L. Kirschstein NRSA T32 CA193201 (to J.H.A.) and NCI F30CA216921 (to K.M.W.). Z.L. is a Hollings Cancer Center Postdoctoral Fellow. The Conrad Prebys Center for Chemical Genomics wishes to acknowledge the NIH Roadmap Grant U54 HG005033 for providing funds to during its participation as a comprehensive screening center of the Molecular Libraries Probe Production Centers Network (MLPCN). The content of this study is solely the responsibility of the authors and does not necessarily represent the official views of the NIH.

## Author contributions
E.H.S. and D.B.D. synthesized the COH000 prep that was used for structure determination. Structural experiments, including crystallization, X-ray data collection/processing, model building/refinement, and structural analyses, were performed by Z.L., L.Y., J.H.A., K.M.W., and S.K.O. Z.L., L.Y., and R.V. conducted biochemical assays. The manuscript was written by S.K.O. with input and editing from Y.C. and C.D.

## Additional information

**Competing interests:** Y.C. owns equity in SUMO Biosciences, Inc. The other authors declare no competing interests.

