## [Peer Review File · Nature Communications]

Reviewers' comments:

Reviewer #1 (Remarks to the Author):

I found the manuscript "Structure of SUMO E1/allosteric inhibitor complex reveals a novel strategy for targeting ubiquitin and ubiquitin-like modifier signaling" by Olsen and co-workers is very interesting and valuable. The outcome of the study is novel and of interest to the large community of the Ub science.

Modifications by Ubiquitin (Ub) and / or Ub-like regulate essentially all cellular processes in eukaryotes. Therefore, modulation of these modifications is valuable for academic (understanding mechanisms) and therapeutic. For therapeutic purposes it would be of benefit to modulate more specific enzymes like E3s. Nevertheless, E1s inhibitors are important as they provide us a deep understanding of these highly complex enzymes and their mechanisms. Moreover, the therapeutic interest of NEDD8 E1 inhibitor cannot be neglected. The current manuscript provides a new class of inhibitor that binds a conserved pocket composed from some non-conserved residues proximal to the ATP binding site. The non-conserved residues allowed the identification of an irreversible-inhibitor that specifically binds to SUMO-E1 but not to other tested E1s. In the manuscript, the crystal structure of the enzyme in complex with the inhibitor is described in details and compared with other structures of E1 enzymes. The authors provide an accompanying manuscript that describes the finding and some biochemical and in vivo characterization of the new inhibitor.

Major comments:

I suggest to use 'irreversible inhibitor' instead of 'allosteric inhibitor' in the title and most of the text. Irreversible inhibitors bind covalently (or very tightly in a noncovalent manner) hence destroy a functional element of an enzyme as described in the current manuscript. The term "allosteric" is derived from the Greek word 'allos' = other and 'stereos' = "shape". Indeed 'other shape' is seen in the structure as described in the manuscript. However, allosteric compounds also present biochemical properties that were not examined in this manuscript. Allosteric inhibition presents specific cooperative conformational changes depend on variations in the structural stability of different parts of a protein and provides a typical sigmoidal shape of inhibited enzymatic activity. It is certainly possible that COH000 functions in allosteric manner however, that was not examined in the manuscript. Instead of asking the authors to perform series of highly complex biochemical assays to assess if COH000 is allosteric inhibitor, I suggest to change the terminology to 'irreversible inhibitor' and conformational changes or other more accurate description. Moreover, the authors are welcome to discuss the possible allosteric function of the inhibitor they identified in the discussion section.

Page 6 line 109 – The description of the accompanying manuscript (Ref. 41) is insufficient for the general readers of Nature Comm. For example, the principle approach of the HTS procedure is not provided. The authors should add few more lines to better describe the ref-41 manuscript. This is important as there is a gap of knowledge for the readers that must be bridges and there is no guarantee that the accompanying manuscript will be available for the readers while the current manuscript will be published.

Fig. 1A and 1B.

It takes some efforts for a crystallographer to actually see the rotation of the cyan colored domain (SCCH). It worth adding the rotation axis

Page 8 lines 139-140 actually the referred figs (Sup Fig 1 a, b) do not show the packing. The authors can prepare additional sup figure that will show the packing at the relevant interface.

Fig. 2 A Showing a difference omit map (Fo-Fc) will be much more convincing than the 2Fo-Fc map. Specifically, I suggest to show together (with different colors) 2Fo-Fc map for the binding site residues and Fo-Fc difference omit map for the inhibitor. I believe the sigma level for the omit map will be much higher than the 2Fo-Fc map.

Fig 2C and 2D – the gels are convincing and clean. However, this is below the standard of Nature Comm to have these kind of data without quantification. One simple way is to repeat the experiment several times and to quantify the bands. Then show representative gels and the quantified data. I would be happy to hear the other referees' opinion on that, as I mentioned that the gels are convincing and I am not happy to be a tyranny referee. However, it is bothersome that the entire manuscript lacks such quantified data.

Fig. 5C it would be useful to add a panel showing all the interactions of the residues of the determined complex (only) in sticks model with the same orientation to the other surface representations. The size should be identical to the panels of the surfaces. This will facilitate the eye view for all the minor differences in the rest structures

Minor comments:

Abstract- the authors wrote:

1. "...disassembly of the active site and a 180° rotation of the catalytic cysteine-containing SCCH domain..."

This is not clear – rotation of 180 degrees with regard to what? Or compare with what?

This must be improved and clarified.

2. I suggest to not use negative wording "Together our study NOT ONLY provides... but to say it in positive fashion our study provide x and y

3. Page 3 line 49 add 'proteins that alter their function or location'

4. Page 4 line 70 should start with 'The catalytic Cys domain'

5. Page 8 line 144 is not clear.

6. Page 4 line 75 The E1 enzymes for all UbIs share conserved structural features...

Recently, E1 enzyme that utilizes NAD instead of ATP and ribosylation of Arg42 instead of the C-ter of Ub has been described. This could be mentioned here or say 'most' rather than 'all'

7. Lines 82-83 – 130 degrees "closer" - is unclear. 130 degrees compare to what / with respect to what?

Reviewer #2 (Remarks to the Author):

This manuscript reports crystal structures of SUMO E1 alone and covalently bound to an allosteric inhibitor (COH000). The authors have identified a potent inhibitor of SUMOylation through a high-throughput screen and subsequently optimized to obtain COH000 and demonstrated that COH000 covalently linked to Uba2's Cys30 to inhibit SUMO E1's activity in a separate manuscript submitted elsewhere. This manuscript reveals the structure and the mechanism of inhibition. The authors showed that COH000 covalently linked to Uba2's Cys30 and bound to a cryptic site on Uba2's adenylation domain near SUMO E1's adenylation active site. This binding configuration displaces

Uba2's structural features that are critical for the assembly of adenylation active site including ATP binding and stabilization of SUMO's C-terminus. Furthermore, there is a significant rotation in the Uba2's catalytic cysteine domain such that the catalytic cysteine residue approaches the adenylation active site but is displaced by 12Å and hence unable to perform the thioester reaction. Collectively this work reveals a novel inhibitory mechanism of a SUMO E1 inhibitor where it binds to a cryptic site and disrupts catalytic features important for the two-step E1 reaction. This is a beautiful work and is suitable for publication.

Reviewer #1 (Remarks to the Author):

I found the manuscript “Structure of SUMO E1/allosteric inhibitor complex reveals a novel strategy for targeting ubiquitin and ubiquitin-like modifier signaling” by Olsen and co-workers is very interesting and valuable. The outcome of the study is novel and of interest to the large community of the Ub science.

Modifications by Ubiquitin (Ub) and / or Ub-like regulate essentially all cellular processes in eukaryotes. Therefore, modulation of these modifications is valuable for academic (understanding mechanisms) and therapeutic. For therapeutic purposes it would be of benefit to modulate more specific enzymes like E3s. Nevertheless, E1s inhibitors are important as they provide us a deep understanding of these highly complex enzymes and their mechanisms. Moreover, the therapeutic interest of NEDD8 E1 inhibitor cannot be neglected. The current manuscript provides a new class of inhibitor that binds a conserved pocket composed from some non-conserved residues proximal to the ATP binding site. The non-conserved residues allowed the identification of an irreversible-inhibitor that specifically binds to SUMO-E1 but not to other tested E1s. In the manuscript, the crystal structure of the enzyme in complex with the inhibitor is described in details and compared with other structures of E1 enzymes. The authors provide an accompanying manuscript that describes the finding and some biochemical and in vivo characterization of the new inhibitor.

We are grateful to the reviewer for their positive comments and thoughtful analysis of our study.

Major comments:

I suggest to use “irreversible inhibitor” instead of “allosteric inhibitor” in the title and most of the text. Irreversible inhibitors bind covalently (or very tightly in a noncovalent manner) hence destroy a functional element of an enzyme as described in the current manuscript. The term “allosteric” is derived from the Greek word “allos” = other and “stereos” =shape. Indeed “other shape” is seen in the structure as described in the manuscript. However, allosteric compounds also present biochemical properties that were not examined in this manuscript. Allosteric inhibition presents specific cooperative conformational changes depend on variations in the structural stability of different parts of a protein and provides a typical sigmoidal shape of inhibited enzymatic activity. It is certainly possible that COH000 functions in allosteric manner however, that was not examined in the manuscript. Instead of asking the authors to perform series of highly complex biochemical assays to assess if COH000 is allosteric inhibitor, I suggest to change the terminology to “irreversible inhibitor” and conformational changes or other more accurate description. Moreover, the authors are welcome to discuss the possible allosteric function of the inhibitor they identified in the discussion section.

We thank the reviewer for highlighting this issue and it stimulated much discussion among the authors. As noted by the reviewer, the definition of allostery is a ligand that binds to a site distinct from the active site of the protein, resulting in modulation of the activity of that protein, often through conformational changes. And in the context of that widely accepted definition of allostery, we believe that COH000 can be defined as an allosteric inhibitor because in the crystal structure it binds at site distinct from the ATP-binding site and its binding inhibits the activity of E1. By stating that allostery requires demonstration of sigmoidal kinetics, the reviewer is applying a stricter definition of the term, but we would argue that under the broader definition, this is not necessary to consider COH000 as allosteric in its mechanism of action (MOA). In our view, to replace the term “allosteric” with “irreversible” would weaken the message of the manuscript because this does not articulate that COH000 binds at a site distinct from the active site.

We do agree, however, that it also important to convey that the reaction is irreversible, but prefer to use the word “covalent” instead because it implies both irreversibility and also explains that the reaction is covalent. Hence, after much discussion and consideration, we now refer to COH000 as a ‘covalent allosteric’ inhibitor in the title and abstract, as this is more accurate than our initial description of it as simply an allosteric inhibitor. We also explain in the revised manuscript how we define allostery. We hope the reviewer finds this acceptable. We are grateful to the reviewer for stating that performing a series of complicated biochemical experiments to confirm allostery through enzyme kinetics is beyond the scope of this manuscript, particularly since the covalent nature of the inhibitor would make such analyses very challenging.

Page 6 line 109 : The description of the accompanying manuscript (Ref. 41) is insufficient for the general readers of Nature Comm. For example, the principle approach of the HTS procedure is not provided. The authors should add few more lines to better describe the ref-41 manuscript. This is important as there is a gap of knowledge for the readers that must be bridges and there is no guarantee that the accompanying manuscript will be available for the readers while the current manuscript will be published.

This is an excellent point and we have added several sentences that more thoroughly describe the HTS procedure and major results of the ref-41 manuscript.

Fig. 1A and 1B.

It takes some efforts for a crystallographer to actually see the rotation of the cyan colored domain (SCCH). It worth adding the rotation axis

We thank the reviewer for this suggestion and have identified the rotation axes during transition of the SCCH domain from the open (adenylation active) to closed (thioester bond formation active) and the open to COH000-inhibited conformations using the program DynDom. The rotation axes are now highlighted in Figs. 1a and 1b as colored arrows. We also created a movie that highlights conformational changes in SUMO E1 that accompany COH000 binding (relative to the adenylation active SUMO E1 conformation) as Supplementary Movie 1. This was created by morphing the adenylation active SUMO E1 conformation to the COH000-inhibited conformation over thirty steps using the program Chimera. We acknowledge that this represents only one possible path for this conformational transition but decided to include it to facilitate visualization of these large conformational changes.

Page 8 lines 139-140 actually the referred figs (Sup Fig 1 a, b) do not show the packing. The authors can prepare additional sup figure that will show the packing at the relevant interface.

Supplementary Figure 2 now shows the crystal packing of the SUMO E1^{COH000} and SUMO E1^{APO} structures. Two views of the structures related by a 90-degree rotation about the vertical axis are presented. All symmetry-related molecules within 10 Å are shown.

Fig. 2 A Showing a difference omit map (Fo-Fc) will be much more convincing than the 2Fo-Fc map. Specifically, I suggest to show together (with different colors) 2Fo-Fc map for the binding site residues and Fo-Fc difference omit map for the inhibitor. I believe the sigma level for the omit map will be much higher than the 2Fo-Fc map.

As suggested, Supplementary Fig. 2 now includes Fo-Fc omit difference electron density for the inhibitor (contoured at 3σ) in green mesh and 2Fo-Fc electron density (contoured at 1.5σ) for the binding site residues in blue mesh. We have clarified that Fig. 2a includes composite omit map electron density for the inhibitor in the legend and by more clearly labeling the figure. We kept this panel in Fig. 2 because it provides an opportunity to clearly label inhibitor substituents that are discussed throughout the manuscript text.

Fig 2C and 2D : the gels are convincing and clean. However, this is below the standard of Nature Comm to have these kind of data without quantification. One simple way is to repeat the experiment several times and to quantify the bands. Than show representative gels and the quantified data. I would be happy to hear the other referees' opinion on that, as I mentioned that the gels are convincing and I am not happy to be a tyranny referee. However, it is bothersome that the entire manuscript lacks such quantified data.

We appreciate the reviewer's comment and agree that we should quantify the data in these panels. We repeated the experiments presented in Figs. 2c and 2d in triplicate, and data from these three independent experiments were quantitated using densitometry and presented as bar graphs, with error bars representing ± 1 SD.

Fig. 5C it would be useful to add a panel showing all the interactions of the residues of the determined complex (only) in sticks model with the same orientation to the other surface representations. The size should be identical to the panels of the surfaces. This will facilitate the eye view for all the minor differences in the rest structures

We thank the reviewer for this suggestion. In Fig. 5c, we have added a cartoon representation of the SUMO E1^{COH000} complex with the side chains from residues involved in contacts to the inhibitor shown as sticks, in the same orientation and size as the surface representations. To facilitate comparison, we did the same for each of the Ubl E1^{COH000} models. This made panel D from the original figure redundant, so we removed it from the modified version of Fig. 5.

Minor comments:

Abstract- the authors wrote:

1. "disassembly of the active site and a 180Å° rotation of the catalytic cysteine-containing SCCH domain"

This is not clear: rotation of 180 degrees with regard to what? Or compare with what?

This must be improved and clarified.

We agree that this was ambiguous and have rewritten this sentence to emphasize that the conformational changes we highlight in the SUMO E1^{COH000} structure are relative to the 'adenylation active' conformational state of SUMO E1.

2. I suggest to not use negative wording "Together our study NOT ONLY provides" but to say it in positive fashion our study provide x and y

We have rewritten this sentence such that the two points are stated in a positive fashion.

3. Page 3 line 49 add "proteins that alter their function or location"

We have rewritten this sentence as suggested.

4. Page 4 line 70 should start with "The catalytic Cys domain"

We have rewritten this sentence as suggested.

5. Page 8 line 144 is not clear.

We thank the reviewer for pointing this out and have reworded this sentence in an attempt to improve its clarity.

6. Page 4 line 75 The E1 enzymes for all UbIs share conserved structural features. Recently, E1 enzyme that utilizes NAD instead of ATP and ribosylation of Arg42 instead of the C-ter of Ub has been described. This could be mentioned here or say "most" rather than "all"

We have added a sentence that highlights the recent discovery of an alternative pathway for Ub activation by *Legionella* SidE effector family members.

7. Lines 82-83 130 degrees "closer"- is unclear. 130 degrees compare to what / with respect to what?

We have clarified that the stated 130-degree rotation of the SCCH domain that accompanies thioester bond formation is with respect to the 'open' SCCH domain conformation observed in adenylation active snapshots of E1.

Reviewer #2 (Remarks to the Author):

This manuscript reports crystal structures of SUMO E1 alone and covalently bound to an allosteric inhibitor (COH000). The authors have identified a potent inhibitor of SUMOylation through a high-throughput screen and subsequently optimized to obtain COH000 and demonstrated that COH000 covalently linked to Uba2s Cys30 to inhibit SUMO E1s activity in a separate manuscript submitted elsewhere. This manuscript reveals the structure and the mechanism of inhibition. The authors showed that COH000 covalently linked to Uba2s Cys30 and bound to a cryptic site on Uba2s adenylation domain near SUMO E1s adenylation active site. This binding configuration displaces Uba2s structural features that are critical for the assembly of adenylation active site including ATP binding and stabilization of SUMOs C-terminus. Furthermore, there is a significant rotation in the Uba2s catalytic cysteine domain such that the catalytic cysteine residue approaches the adenylation active site but is displaced by 12A and hence unable to perform the thioester reaction. Collectively this work reveals a novel inhibitory mechanism of a SUMO E1 inhibitor where it binds to a cryptic site and disrupts catalytic features important for the two-step E1 reaction. This is a beautiful work and is suitable for publication.

We thank the reviewer for their enthusiasm regarding our work.

REVIEWERS' COMMENTS:

Reviewer #1 (Remarks to the Author):

The authors addressed all the points an adequate manner.
I therefore find the manuscript ready for publication in N. Comm.